# Mechanisms and Influence of Casing Shear Deformation near the Casing Shoe, Based on MFC Surveys during Multistage Fracturing in Shale Gas Wells in Canada

**Yan Xi [1],\*** ⬡, **Jun Li [1],\*, Gonghui Liu [1,2], Jianping Li [3] and Jiwei Jiang [1]**

[1] College of Petroleum Engineering, China University of Petroleum-Beijing, Beijing 102249, China; lgh1029@163.com (G.L.); jiangjiwei0603@126.com (J.J.)
[2] College of Mechanical Engineering and Applied Electronics Technology, Beijing University of Technology, Beijing 100022, China
[3] CNPC Logging Co., LTD, Xi'an 710000, China; cpl-ljp@cnpc.com.cn
\* Correspondence: garfield0510@163.com (Y.X.); lijun446@vip.163.com (J.L.);
Tel.: +86 010-89731225 (Y.X.); +86 010-89731225 (J.L.)

**Abstract:** Casing shear deformation has become a serious problem in the development of shale gas fields, which is believed to be related to fault slipping caused by multistage fracturing, and the evaluation of the reduction of a casing's inner diameter is key. Although many fault slipping models have been published, most of them have not taken the fluid-solid-heat coupling effect into account, and none of the models could be used to calculate the reduction of a casing's inner diameter. In this paper, a new 3D finite element model was developed to simulate the progress of fault slipping, taking the fluid-solid-heat coupling effect during fracturing into account. For the purpose of increasing calculation accuracy, the elastoplastic constitutive relations of materials were considered, and the solid-shell elements technique was used. The reduction of the casing's inner diameter along the axis was calculated and the calculation results were compared with the measurement results of multi-finger caliper (MFC) surveys. A sensitivity analysis was conducted, and the influences of slip distance, casing internal pressure, thickness of production and intermediate casing, and the mechanical parameters of cement sheath on the reduction of a casing's inner diameter in the deformed segment were analyzed. The numerical analysis results showed that decreasing the slip distance, maintaining high pressure, decreasing the Poisson ratio of cement sheath, and increasing casing thickness were beneficial to protect the integrity of the casing. The numerical simulation results were verified by comparison to the shape of MFC measurement results, and had an accuracy up to 90.17%. Results from this study are expected to provide a better understanding of casing shear deformation, and a prediction method of a casing's inner diameter after fault slipping in multistage fracturing wells.

**Keywords:** multistage fracturing; shear deformation; numerical simulation; fluid-solid-heat coupling; Multi Finger Caliper

## 1. Introduction

Horizontal well and multistage fracturing are the two key techniques for shale gas development, by creating complex fracture networks within tight formations [1–5]. Due to the fact that tens of thousands of cubic meters of fracturing fluid are pumped into the downholes of shale gas wells and injected into the matrix of shale reservoirs, geostatic stress can be changed due to the elastic response

of the rock mass to hydraulic fracturing. Pore pressure can also be changed due to fluid diffusion along a permeable fault zone [6]. As a result of this, the casing string is exposed to a complex mechanical environment in the downhole, and therefore the risk of casing deformation increases dramatically [7–9]. Previous studies have shown that casing deformations were observed during multistage fracturing in the United States and China, where the commercial development of shale gas is underway and investment is on the rise [10,11]. Casing deformation has created a lot of problems for shale gas well completion and development, for example, bridge plugs could not be run to the projected depth, and normal stimulation operations were unable to be carried out. As a result of this, some fracturing sections either needed to be repaired, which increased the cost of well completion, or could only be abandoned, which decreased the productivity of the shale gas wells. In addition, based on previous studies about casing deformation in conventional oil and gas wells, it is known that casing deformation always intensifies over time, which could lead to security issues for the public or the shut-in of the well. Therefore, there is an urgent need to analyze the mechanisms and propose effective solutions for casing deformation.

Multi-finger calipers (MFC) and lead molds are two effective tools for monitoring the deformation features of the deformed part of a casing [12,13]. Based on investigations into casing deformation by using both tools in China, there were four different types of casing deformations that were delineated, including extrusion deformation, shear deformation, bending deformation, and buckling. Casing shear deformation represented the largest portion of all of the deformed points. Statistic data showed that: (a) up to March 2016, a total of 90 horizontal wells were successfully fractured in Weiyuan-Changning block, Sichuan basin, casing deformation occurred in 32 wells, and 47 deformed points were found [14,15]. 61.7% of all the casing deformed points were examplse of shear deformation [16], and (b) by the end of May 2018, six horizontal wells were successfully fractured in Weirong block, where casing deformation occurred in five wells and 17 deformed points were found. Most of the deformed points were due to shear deformation. Serious casing deformation has occurred during multistage fracturing in shale gas wells in Simonette, Canada, although there is no previous public data. MFC surveys were conducted in five pads, including 28 wells in this study, and the statistical data showed that 52.2% of all the deformed points were due to shear deformation. From the above data, it can be seen that the study of the mechanisms of casing shear deformation appears to be particularly important to the research on casing deformation during multistage fracturing in shale gas wells.

Many related studies have been carried out, but most of them emphasized the inducement of casing shear deformation [17]. Based on the previous research work, faults were easily reactivated when fracturing fluid flew into the cracks in the formation, and were more likely to slip along the unstable bedding planes or natural fractures under the action of their own gravity or external forces [18,19]. Then, the casing strings which passed through the faults were sheared. Qian et al. [20] pointed out that formation stress changed, due to opened and propped hydraulic fractures, and caused natural fractures to open or slip, which increased the risk of casing shear deformation. According to the microseismic data, Zoback and Snee [21] believed that the high pore pressure generated during hydraulic fracturing operations induced slip on preexisting fractures and faults with a wide range of orientations. Meyer et al. [11] analyzed the seismic data and multi-finger caliper data, and suggested that shear failure of pre-existing faults was likely the main cause of casing deformation. Some scholars have proposed a similar viewpoint and complimented the study, pointing out that when the borehole trajectory was inclined upward along the formation, after fracturing fluid flowed into the shale beddings, the fault could slip because of the effect of gravity [15,16]. The mechanisms of fault slippage during or after multistage fracturing were discussed in most of the current research, but few have calculated the variation of the casing's inner diameter, which was the determining factor of whether the bridge plug could pass through the deformed part of the casing, and therefore should be the evaluation basis for effective solutions to address the problem of casing shear deformation.

Analysts can more precisely identify the deformation situation of a casing's deformed parts by using an MFC tool. Some scholars have expounded upon the application of an MFC tool in conventional oil and gas wells, which indicated that MFC data could accurately reflect the actual deformation of the casing string in the well [12,13,22]. Despite the clear benefits of MFC surveys, this kind of technique remains challenging to implement in extensive regional oil and gas fields, due to the significant cost of a full system. As a result, there has been little research providing the evidence of MFC data, and only some conducted prospective studies. Marc [22] collected MFC measurement results from 30 wells from 2003 to 2013, showing that the shear deformation features were localized over a relatively short length (several feet), and resulted from a relative displacement of the upper part of the well compared to the lower part. According to the measurement results of the MFC tool, some mathematical and numerical models were established to simulate the progress of fault slipping, so as to calculate the degree of casing shear deformation. Gao et al. [23] analyzed the characteristics of casing shear deformation, established a 3D finite element model to simulate the stress-strain status and the deformation process, and pointed out that no cementing near the position of slipping was beneficial to the mitigation of casing shear deformation. Chen et al. [16] presented a mathematical model for establishing the relationship between microseismic moment magnitude and degree of casing sheared deformation. Guo et al. [24] developed a numerical model and calculated the influences of slip distance, slip angle, and the mechanical parameters of cement sheath on casing stress. But few of the above studies analyzed the difference between measurement results and actual deformation, and have not put forward solutions to casing shear deformation and showed their validity in engineering in practice. Therefore, it is still a great challenge to solve the problem of casing shear deformation.

In this study, MFC surveys on casing deformation in Canada were implemented, and the mechanisms of casing shear deformation that occurred at the interface between different layers were studied. A new 3D numerical model was developed to simulate the progress of fault slipping, the variation of a casing's inner diameter along the axis was calculated based on the analysis of MFC surveys. The numerical simulation results were verified through the measured data. Six influential factors, including the slip distance, casing inner pressure, thickness of production casing and intermediate casing, and mechanical parameters of cement sheath were analyzed. Furthermore, proposals for mitigation or avoidance of casing shear deformation were suggested, and some of them were applied to the engineering in practice and were proven to be effective.

## 2. Overview of Casing Shear Deformation in Simonette

### 2.1. Field Description

Simonette is one of the most important shale gas blocks in Canada, and is located in the west of Alberta Basin. The shale reservoir is situated in the Upper Devonian Duvernay Formation (Woodbend Group), which is composed of multicyclic units of black organic-rich shale and bituminous carbonates, ranging between 25 and 60 m in thickness [25], and extends throughout most of central Alberta. The vertical depth of the reservoir is approximately 3800–3950 m, and the Duvernay Formation exhibits a westward increase in thermal maturity from the immature to the gas-generation zone [26]. Much of the Duvernay Formation in the area of optimal maturity is overpressured, with noted cases beyond gradients of 1.9 MPa/100m in deep basin settings [25,27]. And the temperature gradient is approximately 3.3–3.7 °C/100 m.

Figure 1 displays the typical well architecture deployed for the development of Simonette. The main drilling phases are: (1) a 349 mm section drilled from the wellhead to approximately 620 m; (2) a 222 mm section drilled from the previous shoe down to the Ireton formation (the true vertical depth (TVD) is approximately 3750 m), including the vertical section and part of indication section; (3) a 171 mm section drilled from the previous shoe down to the Duvernay formation (the TVD is approximately 3885 m), including part of indication section and the whole horizontal section.

The depth of the interface between the Nisku and Ireton formations is about 3742 m. It is worth noting that this interface happens to cross-cut the intermediate casing, which is close to the casing shoe.

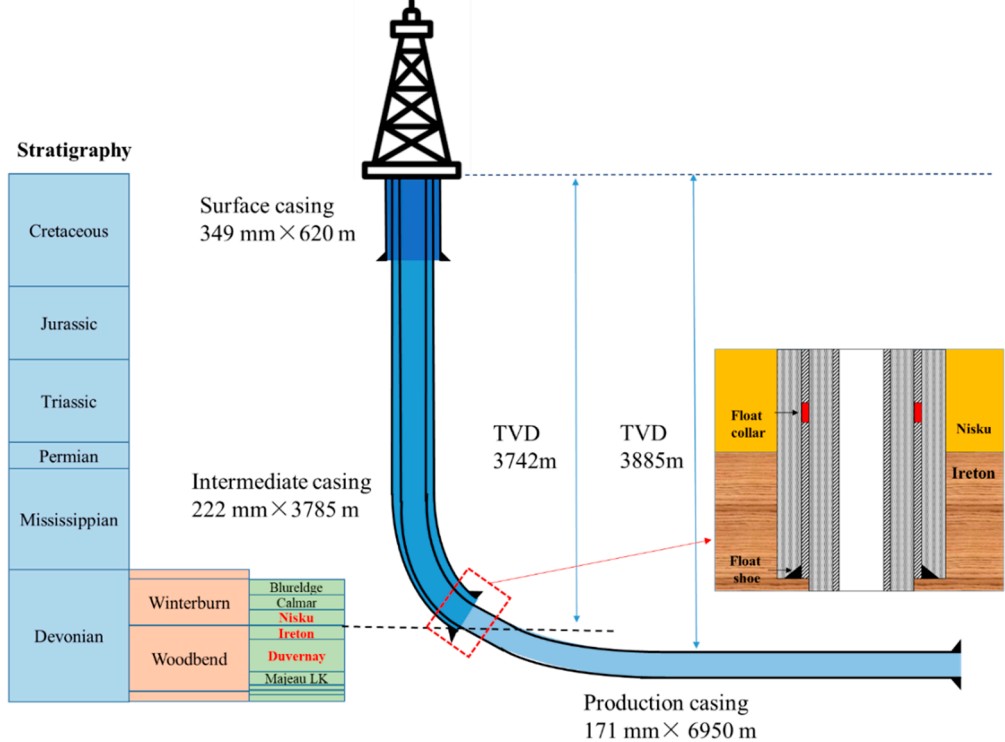

**Figure 1.** Geological stratification and well structure.

## 2.2. Casing Shear Deformation

Casing perforation completion and bridge plug staged fracturing were adopted in shale gas wells in Simonette. The number of fracturing segments was approximately 40, each stage was injected with about 1500 m$^3$ of fracturing fluid, with a displacement of 12–14 m$^3$/min and pumping pressure of over 70 MPa. 5 pads including 28 wells were investigated by MFC surveys. Casing deformation occurred in 16 wells during multistage fracturing. 23 deformed points were found, and there were five different types of deformed points, including extrusion deformation, shear deformation, bending deformation, buckling, and casing holes, as shown in Figure 2. Statistical data showed that 52.2% of all of the deformed points were shear deformation.

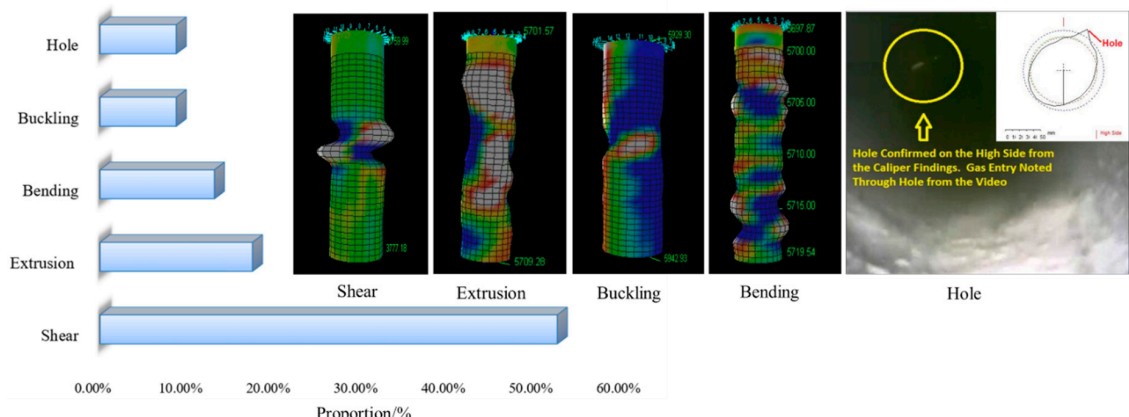

**Figure 2.** Five types of casing deformation and their respective proportions.

Furthermore, the shear deformed points can be classified into two types according to the positions of occurrence: (a) the first type of shear deformed points was located at the position of the interface between the Nisku and Ireton formations, and accounted for 75% of the total shear deformed points; (b) the second type of shear deformed points was located in the horizontal segment, and accounted for 25%. Therefore, it is very meaningful to clarify the mechanism of the first type of casing shear deformation, which is also the aim of this study.

From the introduction above, it is known that the production and intermediate casings were cross-cut by the interface between the Ireton and Nisku formations. According to the statistical data about the depth of the first type of casing shear deformation, all of the shear deformed points occurred at the interface and in the bottom production casing above the casing shoe, as shown in Figure 3. In addition, the lengths of the deformed parts were relatively short, about 1.2–2 m.

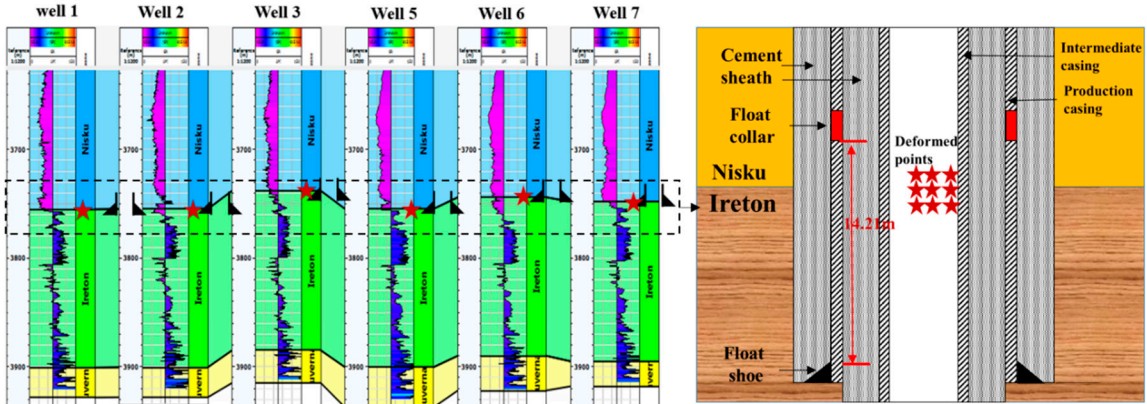

**Figure 3.** Locations of the first type of casing shear deformed points.

*2.3. Difference Between Measurement Results and Actual Shear Deformation*

MFC data can be used to assess the inner conditions of the casing after deformation. As a result of this, shear deformation features identified in the casing turned out to be critical. During the research process, most scholars considered an MFC tool that was centered in the casing, and ignored the transition from measurement results to actual deformation morphology. Indeed, when the MFC tool string runs through the deformation position in the casing, the two centralizers force the middle of the tool string (including the caliper tool) off-center (Figure 4a), which affects the measurement data, and therefore the 3D morphology reflected by the rough data is incorrect.

The difference between the shapes reflected from rough measurement results and actual deformation is illustrated in Figure 4b. From the 3D views of the rough data, it can be seen that the casing appears to be sheared in the upper and lower portions, but the reality shown after data transition indicates that the casing was sheared only at one side. This illustrates that negligence of this condition can lead to wholly misleading results. The deviation of the casing was defined as the degree of casing shear deformation, and statistical data showed the degree had already reached almost 45 mm. The deformation degree (Figure 4b) can be used to describe the slip distance to some extent, but can not be used to measure the reduction of the casing's inner diameter. Therefore, the deformation degree can not be used as the basis of evaluating whether the bridge plug could pass through the deformed part. As a result of this, the relationship between slip distance and the reduction of the casing's inner diameter should be established.

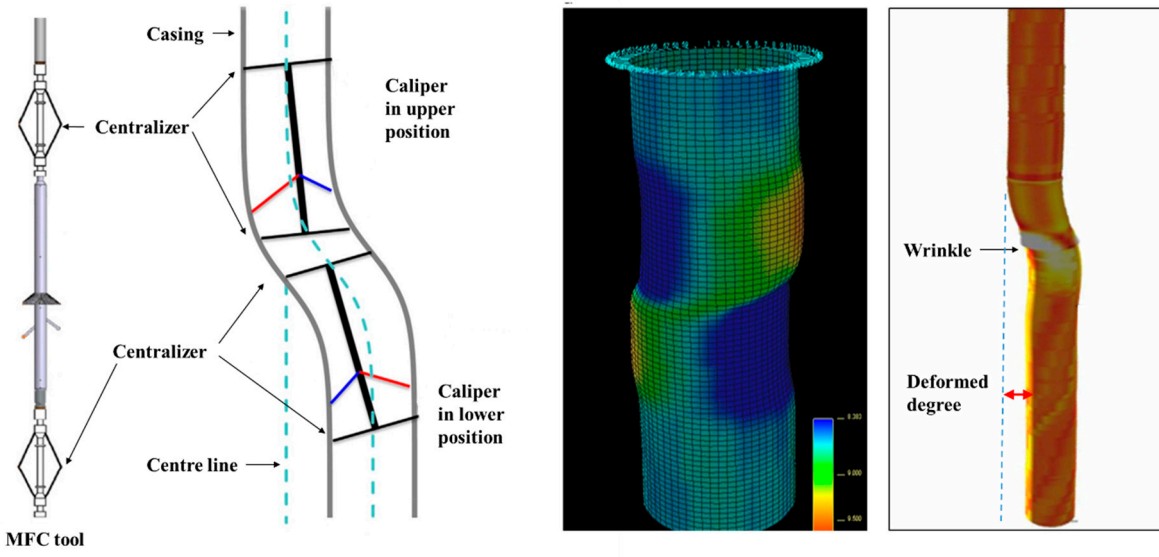

(**a**) Schematic diagram　　　　　　　　　(**b**) Measurement results and actual conditions

**Figure 4.** Schematic diagram and comparison of measurement results and actual conditions.

## 3. Mechanisms of Casing Shear Deformation induced by Multistage Fracturing

### 3.1. Mechanisms of Casing Shear Deformation

Previous studies have proven that fault slipping was the main reason for casing shear deformation. On the basis of worldwide observations, casing shear deformation caused by fault slipping can be classified into three main categories:

(1) Shear deformed points overlap with natural fractures (type 1, Figure 5). This is the most common mechanism of casing shear deformation, and was identified in almost all of the shale gas fields [14–17,21]. This type of casing shear deformation is not necessarily restricted to the high shear stress area, and the position of the shear deformed points are more directly related to the location and orientation of natural fractures, rather than the location of the crustal stress.

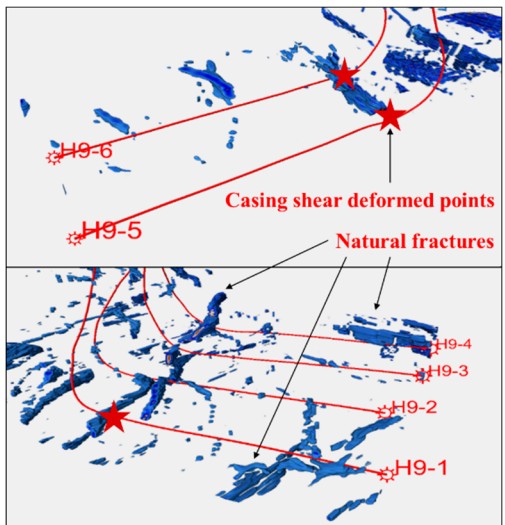

**Figure 5.** Deformed points overlap with natural fractures.

(2) Shear deformed points appeared near the landing point (type 2, Figure 6). The landing point is the starting point of the horizontal segment. Nearly all the trajectories of the wells which incurred

casing deformation of this type were inclined upward along the formation. Some scholars [14,21] have believed that during the progress of fault slipping, the gravity of the fault plays an important role in activating the faults after the fracturing fluid flew into the interface of bedding planes.

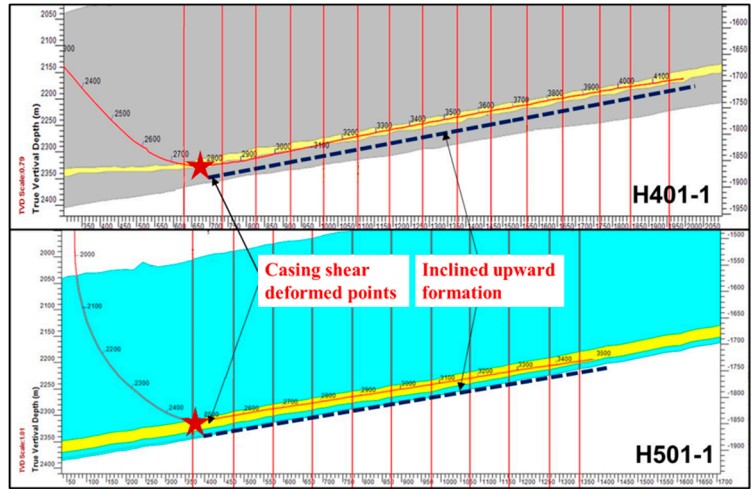

**Figure 6.** Deformed points near the landing point.

(3) Shear deformed points at the interface between different layers (type 3). Shear deformed points located at the position of the interface between the Nisku and Ireton formations are of this type. The interface between different formations is opened, due to the high bottom hole pressure during the operation of well cementation, and the friction coefficient between the formations decreases dramatically. During the progress of multistage fracturing, the fault below the interface is easily activated and it can slip along the interface.

### 3.2. Verification of Fault Slipping

Suppose that the interface between Nisku and Ireton formations was a weak interface, especially after the interface was opened and the cement slurry flowed into it. The shear and normal stress on the oriented plane can be calculated by coordinate transformation. The formation/fault would slip when

$$\tau > f_n \sigma_n + S, \tag{1}$$

where $\tau$ represents the shear stress applied to the fault (MPa); $f_n$ represents the coefficient of friction, dimensionless. And according to Zoback [28], the value range of $f_n$ is from 0.6 to 1. $\sigma_n$ represents the effective normal stress (MPa); $S$ represents the rock cohesive strength (MPa).

Based on Biot's law [29], $\sigma_n$ can be expressed as

$$\sigma_n = S_n - P_p, \tag{2}$$

where $P_p$ is the pore pressure in MPa, and $S_n$ is the normal stress perpendicular to the interface.

The pore pressure increases at the bedding plane during fracturing; when it meets the formula requirements, the minimum increment of pore pressure (MPa) is [28]:

$$P' = \frac{S}{f_n} + \sigma_H + (\sigma_H - \sigma_v)\left(\sin^2 \psi - \frac{\sin \psi \cos \psi}{f_n}\right), \tag{3}$$

where $\Delta\rho_w$ is equal to the yield density in (g/cm$^3$). $\psi$ is the angle between the interface and the maximum horizontal principal stress (degree). The normal stress has been resolved into horizontal ($\sigma_H$) and vertical ($\sigma_v$) components.

The results calculated according to the conditions given in this study (Section 4.3) showed that the critical pore pressure was 81.2 MPa. In addition, the downhole pressure was as high as 115 MPa, indicating that the fault was likely to slip at excessive pore pressure.

In order to demonstrate this further, a microseismic technique was used to monitor the fault slipping which occurred at the position of the interface between the Nisku and Ireton formations [30–33]. From the depth view of 9-31 well, it can be seen that hydraulic fractures already extended to the interface (Figure 7a). Some microseismic events appeared at different locations, which were far apart from each other, but all along the interface, which indicated that the interface was opened during the process of fracturing, and the fault was likely to slip. Microseismic data of the 12-6 pad showed that events were found at the top of Ireton formation (Figure 7b), which was direct evidence of the fault slipping which occurred at the position of the interface between the Nisku and Ireton formations.

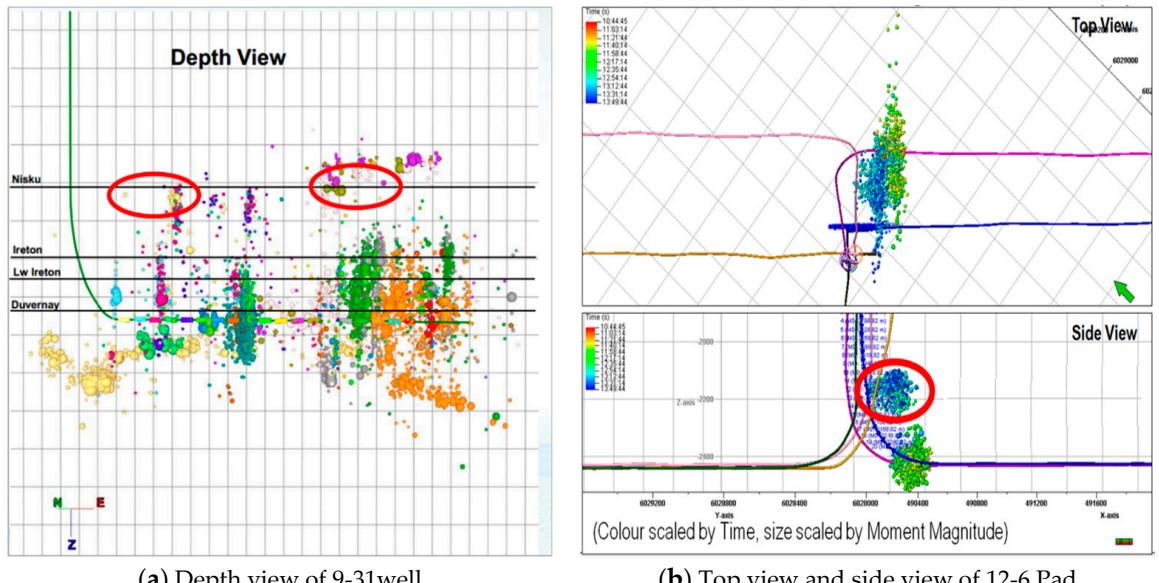

(**a**) Depth view of 9-31well　　　　　　　　(**b**) Top view and side view of 12-6 Pad

**Figure 7.** Microseismic data.

## 4. Numerical Simulation

In order to calculate the variation of the casing's inner diameter, a numerical model was developed and the influential factors were taken into account. Simulations were carried out by using the commercial software Abaqus (6.14-1), which can be used to perform and post-process simulations of various cases, and for statistical sensitivity analysis. The measurement results of the MFC tool are used to confirm the validity of the proposed Finite Element Model (FEM).

### 4.1. Model Geometry and Discretization

Physical model. The assembly, which contains a production casing-cement sheath-intermediate casing -cement sheath formation, located above the intermediate casing shoe, was selected as the research object. It was assumed that the casings were centered and the cement sheaths were integral. The formation includes two blocks, the upper block which represented the Nisku formation was the fixed part, and the lower block which represented Ireton formation was the slip part, as shown in Figure 8a.

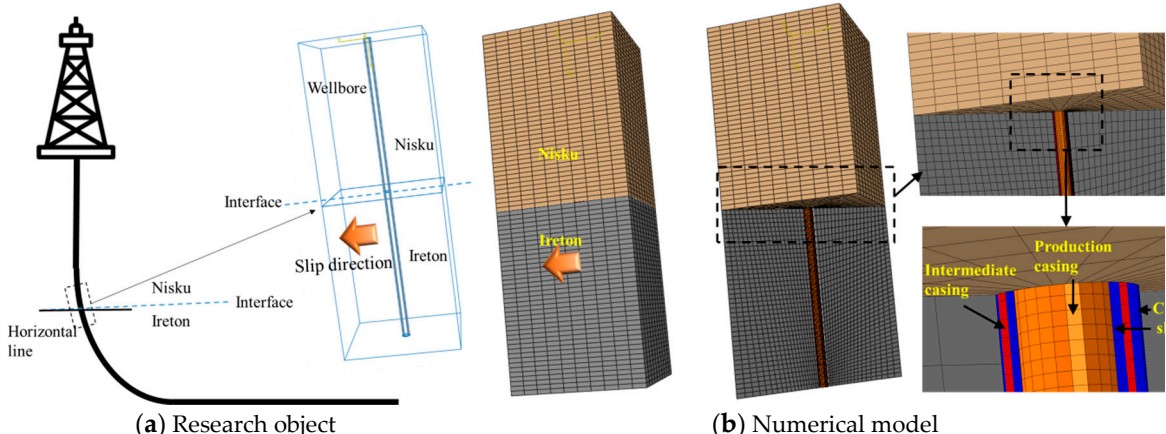

(**a**) Research object           (**b**) Numerical model

**Figure 8.** Research object and numerical model of fault slipping.

Numerical model. A three-dimensional (3D) nonlinear FEM was established to simulate the slip progress of the fault and the mechanical behaviors of the casing shear deformation, as shown in Figure 8b. As assumed above, the casings were centered, and the cement sheaths showed complete integrity. The outer diameter of the model was 3 m × 3 m × 8 m, which was ten times greater than the size of the borehole, thus allowing the influence of the boundary to avoid the effects on the stress.

Discretization. During the simulation, the materials in the model were chosen assuming that the casing followed the elastically-perfect plastic constitutive relationship with the Von Mises yield criterion, and the cement sheath and formation followed the elastic-plastic constitutive relationship with the Mohr-Coulomb criterion. In order to reflect the casing deformation accurately, the solid element (3D stress, C3D8R) is used to analyze the cement sheath-intermediate casing-cement sheath formation, and the shell element (shell, S4R) is used to study production casing. During the progress of discretizing the finite element model, in order to increase the computational accuracy, the structured grid and variable density meshing method is applied to the model.

### 4.2. Boundary Conditions and Simulation Steps

Boundary Conditions. In terms of load and constraint sets, the composite boundary of the upper part was fixed by imposing displacement constraints, and the slip displacement of the Ireton part is imposed on the corresponding formation's surface. Through finite element predefined field function, far-field stress was applied, and the hydraulic pressure was added to the inner wall of the casing. The stress field of the research object, which was part of the inclination segment, could be obtained by three-time rotations from the original coordinate system, and this was then applied to the numerical software (Appendix A). The pressure of fracturing fluid was applied to the casing's inner pressure.

In order to reflect the actual conditions of the casing, the transient temperature change of the model was taken into account. The initial temperature of assembly was equal to formation temperature, and the outside surface of the model was one of the temperature boundaries and was set as a stable thermal source. The casing's inner wall was another temperature boundary, and its temperature was equal to that of the fracturing fluid, which should be calculated by a wellbore temperature field model during fracturing (Appendix B). The variation of time-dependent temperature was the input for the numerical model as a dynamic boundary.

Simulation Steps. At the first step, the temperature variations of the well are calculated when imposing the dynamic boundary at the inner wall of the production casing. Subsequently, the calculated temperature distribution, geo-stress, and the internal pressure are applied to the assembly, and the initial state of equilibrium is reached. Lastly, the lower block slipped and casing deformation occurred, then the displacement of casing's inner wall was calculated. During the third step, for sensitivity purposes, the slip distance, casing inner pressure, the production and intimate

casing thickness, and mechanical parameters of cement sheath will be changed in order to find the optimal parameters during simulation.

*4.3. Geological and Mechanical Parameters*

As a horizontal shale gas well in Simonette, 12-10 wells were drilled to the maximum vertical depth of 3920 m, with a horizontal segment length of 2450 m. The measured and true vertical depths of the interface between the Nisku and Ireton formations were 3786 m and 3742 m, respectively. The horizontal minimum in-situ stress and vertical in-situ stress gradients were 2.3 MPa/100 m, 2.0 MPa/100 m, and 2.5 MPa/100 m, respectively. The hole inclination angle is 24°, and the well azimuth angle is 78°. The pumping pressure of fracturing was 77 MPa, and the fracturing fluid friction is approximately 10 MPa. The fracturing fluid density was 1.28 g/cm$^3$, then the casing's inner pressure in the interface was about 115 MPa. The discharge of fracturing fluid is 12 m$^3$/min, and the fracturing time is 4 h. Other geological and mechanics parameters are shown in Tables 1 and 2.

**Table 1.** Geometric and mechanical parameters of the assembly.

| Component | Outer Diameter (mm) | Young Modulus (GPa) | Poisson's Ratio | Cohesive Strength (MPa) | Internal Friction Angle(°) |
|---|---|---|---|---|---|
| Production Casing | 139.7 | 210 | 0.3 | \ | \ |
| Cement Sheath (2) | 171 | 10 | 0.17 | 8 | 27 |
| Intermediate Casing | 193.7 | 210 | 0.3 | \ | \ |
| Cement Sheath (1) | 222 | 10 | 0.17 | 8 | 27 |
| Formation | \ | 22 | 0.23 | 5 | 39 |

**Table 2.** Thermodynamic Properties of the assembly.

| Materials | Coefficient of Heat Conduction (W·(m·°C)$^{-1}$) | Specific Heat (J·(kg·°C)$^{-1}$) | Density (kg·m$^{-3}$) | Coefficient of Thermal Expansion (10$^{-6}$·°C$^{-1}$) |
|---|---|---|---|---|
| Casing | 45 | 461 | 7800 | 13 |
| Cement sheath | 0.98 | 837 | 3100 | 11 |
| Formation | 1.59 | 1256 | 2600 | 10.5 |

## 5. Results

*5.1. Engineering Verification of the Numerical Simulation*

Figure 9 shows the distribution of casing strain after fault slipping, which indicates that the stain concentrations appear at the position of the slip interface, and shows symmetrical distribution along the center line of the casing. The overall deformed shape of the casing after fault slipping is similar to the actual deformation, as shown in Figure 4b. Line a1-a2 and line b1-b2 are perpendicular, which can represent the casing's inner diameter. Because the plane a1-a2-a'2-a'1 is parallel to the direction of fault slipping, the greatest variation of the casing's inner diameter occurred in this plane after fault slipping. As a result of this, line a1-a'1 and a2-a'2 were selected to accurately assess the variation of the inner diameter along the whole casing.

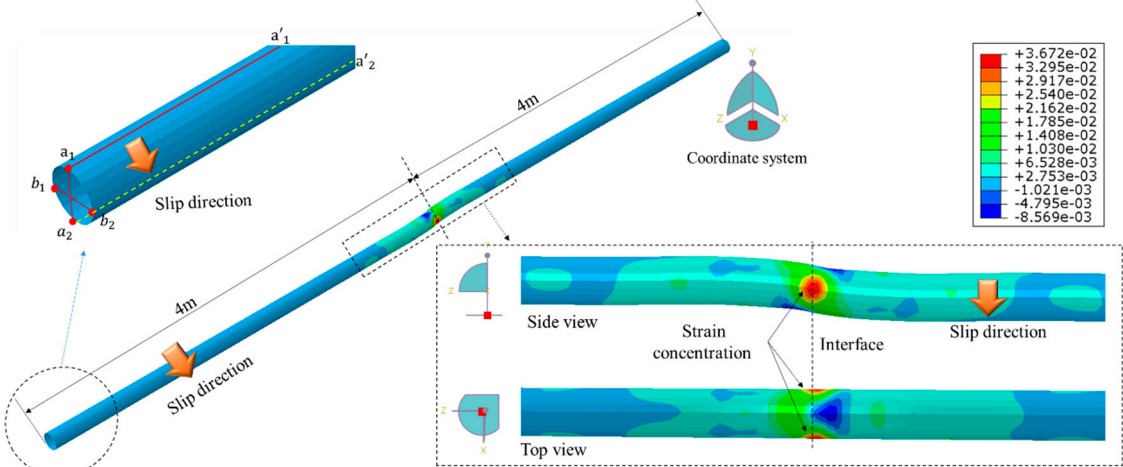

**Figure 9.** Distribution of casing strain (dimensionless).

Figure 10 shows the displacement curves of line a1-a'1 and a2-a'2 after fault slipping. Both curves were divided into three sections, as shown in Figure 10b–d. The displacement differences of the two curves in the first and third part are relatively small, which means that the inner diameter of the casing is almost unchanged. There is a rule for these two parts that the closer the distance from the slip interface, the more obvious the difference between the two lines. The displacement differentiation of the two lines is very obvious in the second part, which demonstrates the casing's inner diameter changes significantly, and structural distortion occurs. Therefore, the second part is the most essential part for deciding whether the bridge plug can pass the deformed part.

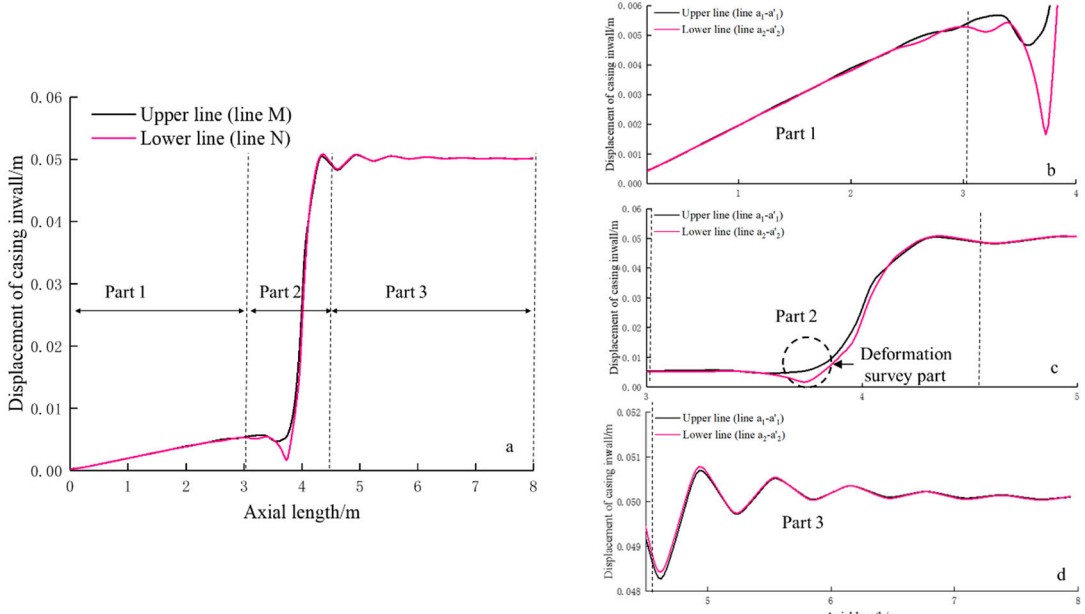

**Figure 10.** Variation of displacement along the casing.

The length of the second part is 1.38 m, which is in accord with the statistical rule. In order to assess the accuracy of the calculation, the simulated and measurement results at the special position are compared, as shown in Figure 11. It is shown that the result of the cross-section at the position of deformed part shows remarkably consistent findings.

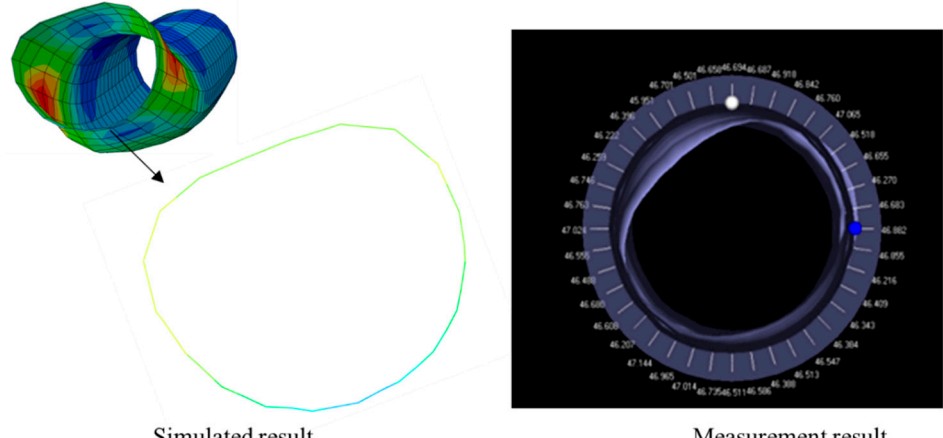

Figure 11. Comparison of the simulation and measurement results.

On the basis of the above analysis, the variation of diameter along the whole casing was calculated, as shown in Figure 12, which indicates that the most dramatic change appears at the position of the slip interface. That the maximum reduction of casing inner diameter is 4.06 mm leads to a drop of 3.35% compared to the original inner diameter.

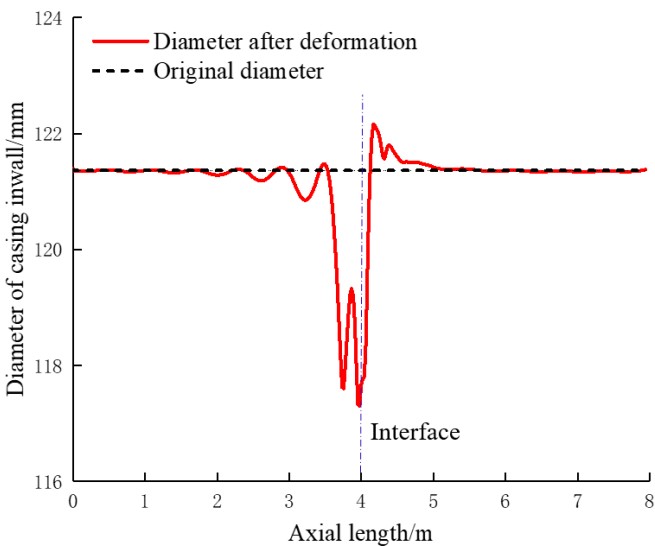

**Figure 12.** Variation of diameter along the casing.

*5.2. Sensitivity Analysis of Casing Shear Deformation*

5.2.1. Influence of Slip Distance

Previous studies have shown that with the increase of slip angle (0–90°), fault slipping had a growing influence on casing shear deformation [16]. Consider the worst-case scenario, which is that the slip angle is 90°, the influence of slip distance on the variation of casing's inner diameter was calculated, as shown in Figure 13. It can be concluded that with the increase of the slip distance, the reduction of the casing's inner diameter increases clearly, and the deformation becomes more and more complex and severe. It is worth noting that the number of concave areas in the curves changes from one to two (Figure 14), which means that the larger slip distance makes it harder for the bridge plug to pass through the deformed part. Also, it is the cause of the appearance of the wrinkle in the inner wall of casing, as shown in Figure 4b.

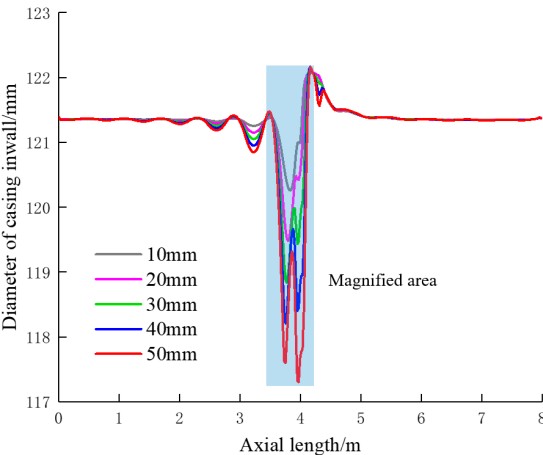

**Figure 13.** Variation of diameter under different slip distances.

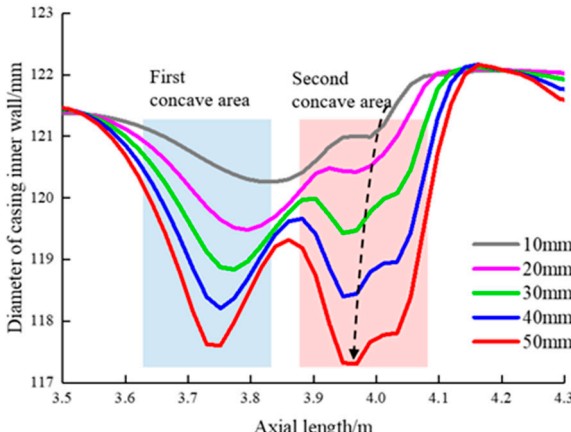

**Figure 14.** Magnified area.

## 5.2.2. Influence of Casing Inner Pressure

Fault slipping could happen during or after fracturing, as the casing's inner pressure is different in different stages. Figures 15 and 16 show the variation of diameter along the casing under different inner pressures. It can be seen that with the increase of inner pressure, the reduction of the casing's inner diameter decreases, which indicates that higher inner pressure is beneficial to maintain the casing's integrity. It should be noted that the casing's inner pressure has little impact on the first concave area in the curves, while it has a relatively significant impact on the second concave area. When the casing's inner pressure is zero, the reduction of diameter is the greatest (4.90 mm), and compared with the reduction when the casing's inner pressure is 115 MPa (the reduction is 4.06 mm), the increase of the diameter reduction is 20.7%. The reason for this is mainly because the casing's inner pressure has a dramatic impact on the equivalent stiffness of the casing string. The higher the casing's inner pressure, the higher the equivalent stiffness.

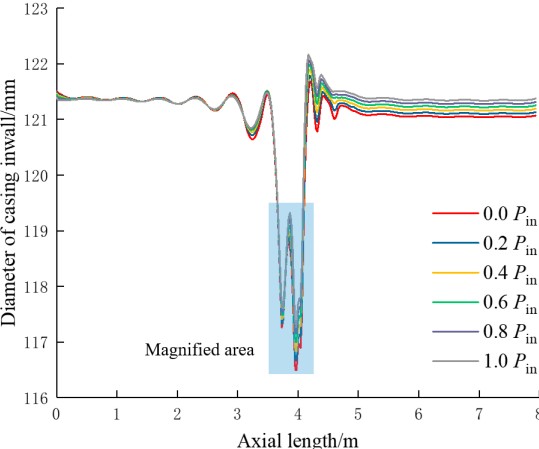

**Figure 15.** Variation of diameter under different casing inner pressures.

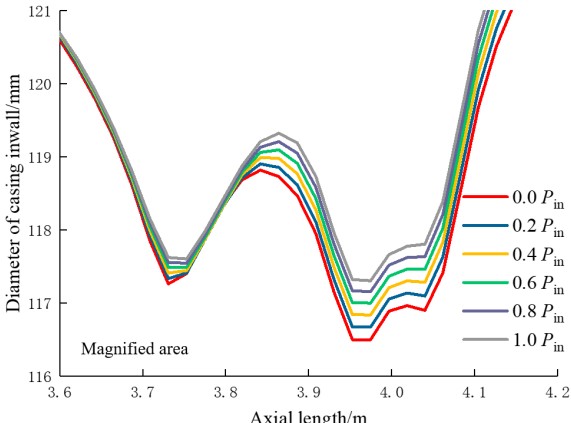

**Figure 16.** Magnified area.

### 5.2.3. Influence of Casing Thickness

With the increase of casing thickness, the shear resistance strength increases. Because the research object contains two casings, including intermediate and production casing, the thickness of both types of casings are changed, so as to evaluate the effectiveness of increasing the casing thickness. Figure 17 shows the variation of the casing's inner diameter under different production casing thicknesses. It can be seen that with the increase of the casing thickness, the reduction of casing's inner diameter is almost unchanged in the first concave area, but decreases clearly in the second concave area (Figure 18). Figure 19 shows the reduction of the casing's inner diameter with the variation of the intermediate casing thickness, and it can be seen that with the increase of the casing thickness, the reductions of the casing's inner diameter decreases nearly linear in both concave areas. Comparing the two methods, increasing the thickness of the intermediate casing above the casing shoe is the more effective.

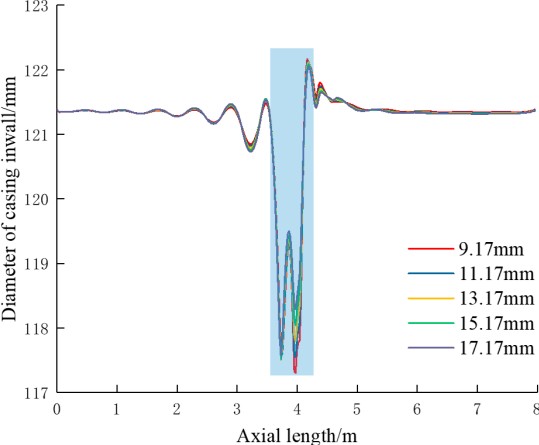

**Figure 17.** Variation of diameter under different production casing thickness.

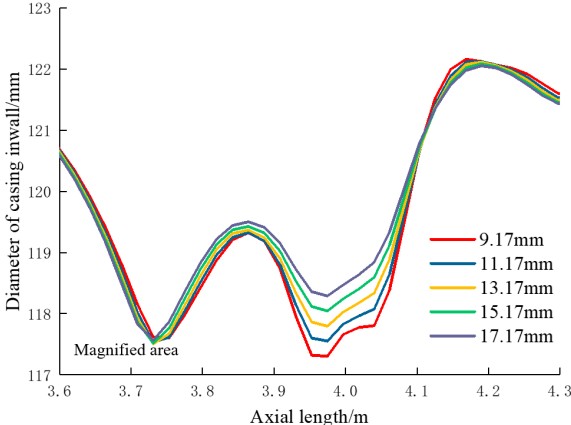

**Figure 18.** Magnified area.

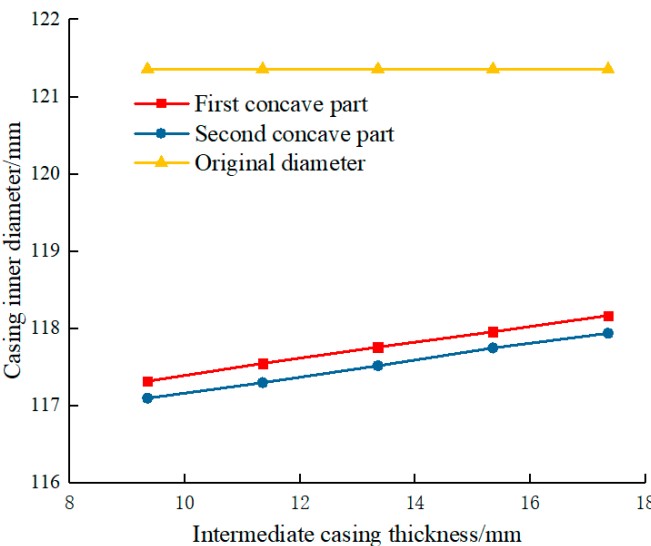

**Figure 19.** Variation of diameter under different intermediate casing thickness.

### 5.2.4. Influence of Cement Sheath Mechanical Parameters

The mechanical parameters of the cement sheath can be adjusted by using different cement slurry formulas, and have an impact on the reduction of the casing's inner diameter. Figure 20 shows the variation of diameter along the casing under different elasticity moduli of the cement sheath. It can be

seen that the influence on the casing's inner diameter in the two concave areas is different. In the first concave area, with the increase of the elasticity modulus of the cement sheath, the reduction of the casing's inner diameter decreases. But in the second concave area, the variation of the diameter has a reverse rule (Figure 21). Figures 22 and 23 show the influence of the Poisson ratio on the casing's inner diameter. It can be seen that with the increase of the Poisson ratio, the reduction of diameter increases, especially in the second concave area, which indicates that the lower Poisson ratio is beneficial for decreasing the reduction of the casing's inner diameter.

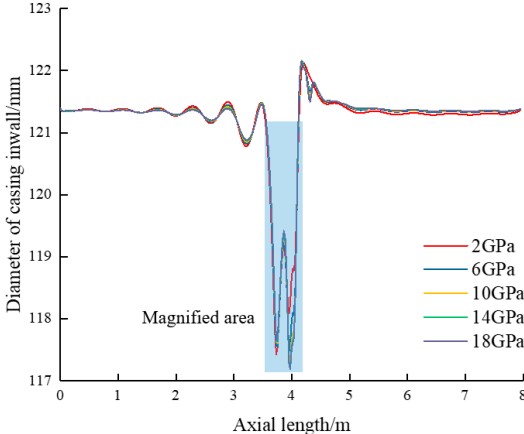

**Figure 20.** Variation of diameter under different elasticity moduli.

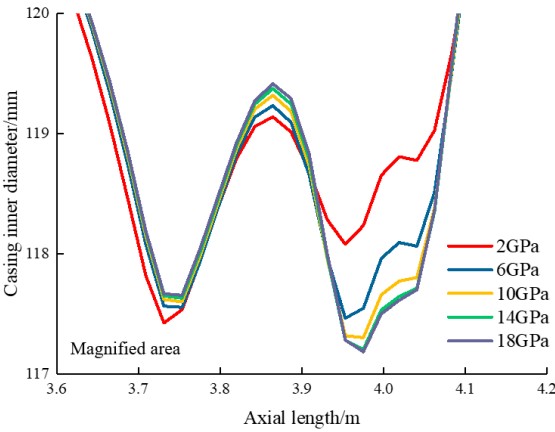

**Figure 21.** Magnified area.

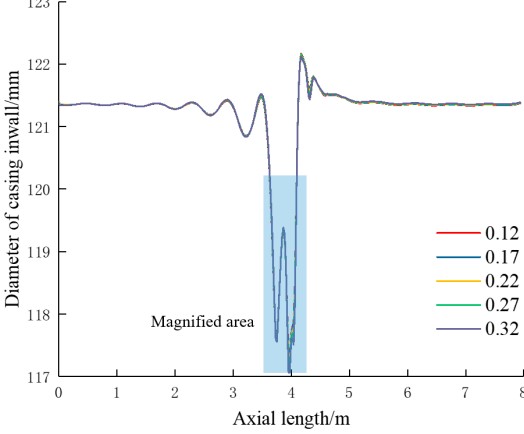

**Figure 22.** Variation of diameter under different Poisson ratios.

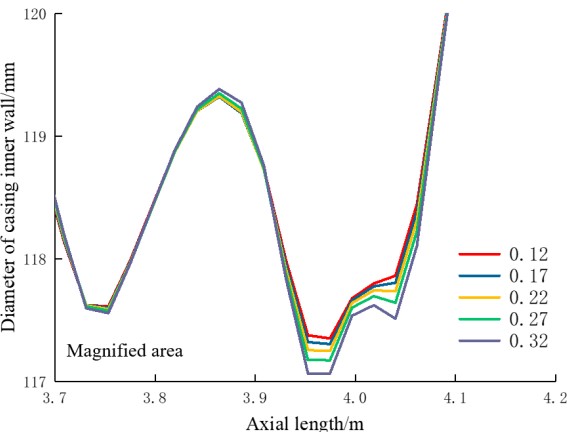

**Figure 23.** Magnified area.

## 6. Results Comparison and Mitigation Method

The method in this study presented a way to evaluate the reduction of casing's inner diameter. Using to the microseismic measurements, microseismic moment magnitude can be obtained. And based on the study of Chen et al. [14], slip distance can be calculated. Then, by using the model proposed in this study, the diameter of a casing's inner wall after fault slipping can be computed. The computed results and the measurement results by using MFC tools were compared, as shown in Figure 24. It can be seen that the numerical method in this study has an accuracy of up to 90.17%, and it can be used to select the soluble bridge plug after casing shear deformation, achieving the purpose of overcoming the problem that MFC measurement is costly, as mentioned above.

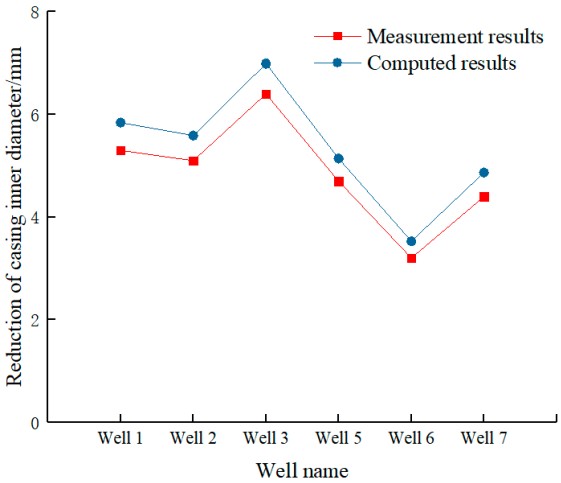

**Figure 24.** Comparison of measurement and computed results.

According to the above analysis, it could be seen that decreasing the slip distance was the best way to protect the casing. In order to mitigate or eliminate fault slipping, the interface between the Nisku and Ireton formations should avoid being opened during the operation of well cementation. As a consequence, the well structure was optimized. The depth of the intermediate casing shoe was reduced to approximately 3600 m, and the depth was more than 100 m above the interface between the Nisku and Ireton formations. The method was applied to nine wells, eight wells did not occur casing shear deformation, which was proved to be the most efficient and economical way. And the engineering practice supported the correctness of the analysis in this study.

## 7. Conclusions

Casing shear deformation occurring during multistage fracturing was monitored by using MFC tools, and the cause of the shear deformed points located at the interface between the Nisku and Ireton formations was analyzed. A new investigation based on the MFC measurement results was carried out, and the impact of influential factors on the reduction of a casing's inner diameter was studied. The following conclusions were drawn:

(1) MFC surveys were carried out to monitor the casing deformation occurring during multistage fracturing in Simonette, Canada. Statistical data showed that shear deformation was the main type of casing deformation, and the shear deformed points can be classified as two types according to the positions of occurrence: shear deformed points located at the position of the interface between the Nisku and Ireton formations (75%), and in the horizontal segment (25%).

(2) The cause of casing shear deformation occurring at the interface between the Nisku and Ireton formations was analyzed. When the interface between the different layers was opened during the operation of well cementation, the friction coefficient between the layers decreases dramatically. During multistage fracturing, the fault was activated and slipped along the interface, which was verified by the microseismic data.

(3) A numerical model has been developed to analyze the reduction of the casing's inner diameter. The simulation results showed that: (a) fault slipping caused the reduction of casing inner diameter, and the maximum change appeared at the position of the interface of the two formations; (b) the cross-section of the casing calculated by the numerical model was similar to the shape reflected by MFC data in that particular position.

(4) A sensitivity analysis was carried out and the influence of slip distance, the casing's inner pressure, the mechanical parameters of the cement sheath, and the intermediate and production casing thickness on the reduction of the casing's inner diameter were analyzed. According to the numerical analysis results, decreasing the slip distance, maintaining high pressure, decreasing the Poisson ratio of the cement sheath, and increasing the casing thickness were beneficial for protecting the integrity of casing. Furthermore, the effectiveness of increasing the intermediate casing thickness is greater than increasing that of the production casing.

(5) Measurement results were compared with computed results to verify the method proposed in this study. The numerical method in this study has an accuracy of up to 90.17%, which can be used as basis for choosing soluble bridge plugs. In addition, the well structure was optimized, and the depth of the intermediate casing shoe was reduced to approximately 3600 m, which was more than 100 m above the interface between the Nisku and Ireton formations. The effectiveness of this method was verified by engineering in practice, as eight of nine wells did not incur casing deformation after implementation of the method.

**Author Contributions:** Y.X. contributed to developing the mathematical model, performed the data analysis, and wrote the manuscript; J.L. (Jun Li) and G.L. supervised the research and edited the manuscript. J.L. (Jianping Li) and J.J. analyzed the engineering data.

**Funding:** "Study on failure mechanisms and control methods of wellbore integrity of shale gas horizontal wells" (U1762211);"Optimum research of non-uniform cluster perforation along the long horizontal section in heterogeneous shale reservoirs" (51674272).

**Acknowledgments:** This research was financially supported by the Key Program of National Natural Science Foundation of China "Study on failure mechanisms and control methods of wellbore integrity of shale gas horizontal wells" (U1762211), the National Natural Science Funds "Optimum research of non-uniform cluster perforation along the long horizontal section in heterogeneous shale reservoirs" (51674272).

**Conflicts of Interest:** The authors declare no conflict of interest.

## Appendix A

The stress field of the formation was usually described by triaxial principal in-situ stresses, including the horizontal maximum in-situ stress, the horizontal minimum in-situ stress and the vertical

in-situ stress ($\sigma_H$, $\sigma_h$ and $\sigma_v$, MPa). Although the stress field in the vertical section of the shale gas well was stay the same with in-situ stress, the mechanical state in the inclination section was different. For the reason that the research object was located in the inclination section, the in-situ stress should be converted to the stress tensor in the wellbore coordinate system.

A coordinate rotation matrix was built in order to transform the original coordinate system to wellbore coordinate system. It could be expressed as

$$L = L_y L_z = \begin{bmatrix} \cos\alpha & 0 & -\sin\alpha \\ 0 & 1 & 0 \\ \sin\alpha & 0 & \sin\alpha \end{bmatrix} \begin{bmatrix} \cos\beta & \sin\beta & 0 \\ -\sin\beta & \cos\beta & 0 \\ 0 & 0 & 1 \end{bmatrix} = \begin{bmatrix} \cos\alpha\cos\beta & \cos\alpha\sin\beta & -\sin\alpha \\ -\sin\beta & \cos\beta & 0 \\ \sin\alpha\cos\beta & \sin\alpha\sin\beta & \cos\alpha \end{bmatrix} \tag{A1}$$

where, $\alpha$ represents the hole inclination angle,°; $\beta$ represents the well azimuth angle,°; $L_y$ and $L_z$ represent the direction cosine matrixes rotating around y-axis and z-axis based on the right-hand rule.

$$L_y = \begin{bmatrix} \cos\alpha & 0 & -\sin\alpha \\ 0 & 1 & 0 \\ \sin\alpha & 0 & \sin\alpha \end{bmatrix} \tag{A2}$$

$$L_z = \begin{bmatrix} \cos\beta & \sin\beta & 0 \\ -\sin\beta & \cos\beta & 0 \\ 0 & 0 & 1 \end{bmatrix} \tag{A3}$$

Then, the stress tensor ($\sigma_{ij}$) in the wellbore coordinate could be calculated by

$$\sigma_{ij} = \begin{bmatrix} \sigma_{xx} & \sigma_{xy} & \sigma_{xz} \\ \sigma_{yx} & \sigma_{yy} & \sigma_{yz} \\ \sigma_{zx} & \sigma_{zy} & \sigma_{zz} \end{bmatrix} = L \begin{bmatrix} \sigma_H & & \\ & \sigma_h & \\ & & \sigma_v \end{bmatrix} L^T \tag{A4}$$

where, $L^T$ represents the transposition of $L$, dimensionless.

## Appendix B

A linear relationship was assumed between the formation temperature and depth:

$$T_z = T_b + \alpha(z - b) \tag{A5}$$

where $T_z$ represents formation temperature at a certain depth, °C, $T_b$ represents land surface temperature, °C, $\alpha$ represents the geothermal gradient, °C/m, $z$ represents reservoir depth, m; $b$ represents the benchmark depth, m.

Based on the law of energy conservation, a model of the wellbore temperature field during fracturing was established. The wellbore was divided into infinitesimal equal parts along the wellbore. The conservation of energy of fluid in wellbore is calculated according to Equations (A6) and (A7):

$$Q\rho_0 C_0 T_{0,j-1}^{n+1} - Q\rho_0 C_0 T_{0,j}^{n+1} + 2\pi r_0 \Delta H_j U \left( T_{1,j}^{n+1} - T_{0,j-\frac{1}{2}}^{n+1} \right) + W_j = \pi r_0^2 \Delta H_j \rho_0 C_0 \frac{T_{0,j-\frac{1}{2}}^{n+1} - T_{0,j-\frac{1}{2}}^n}{\Delta t} \tag{A6}$$

$$W_j = \lambda_{fj} \frac{\Delta H_j}{r_0} \frac{\rho_0}{2} \frac{Q^3}{\pi^2 r_0^2} \tag{A7}$$

$$\frac{T_{0,j}^{n+1} - T_{0,j-1}^{n+1}}{2} = T_{0,j-\frac{1}{2}}^{n+1}$$

The conservation of energy of solid elements with a fluid foundation and for solid elements was shown in Equations (A8) and (A9), respectively:

$$-2\pi r_0 \Delta H_j U\left(T_{1,j}^{n+1} - T_{0,j-\frac{1}{2}}^{n+1}\right) + 2\pi r_1 \Delta H_j K_1 \frac{T_{2,j}^{n+1} - T_{1,j}^{n+1}}{\frac{r_2-r_0}{2}} = \pi\left(r_1^2 - r_0^2\right)\Delta H_j \rho_1 C_1 \frac{T_{1,j}^{n+1} - T_{1,j}^{n}}{\Delta t} \quad \text{(A8)}$$

$$\frac{4\pi r_{i-1}\Delta H_j K_{i-1}}{r_i-r_{i-2}}T_{i-1,j}^{n+1} + \left(-\frac{4\pi r_{i-1}\Delta H_j K_{i-1}}{r_i-r_{i-2}} - \frac{4\pi r_i \Delta H_j K_i}{r_{i+1}-r_{i-1}} - \frac{\pi\left(r_i^2-r_{i-1}^2\right)\Delta H_j \rho_i C_i}{\Delta t}\right)T_{i,j}^{n+1} + \frac{4\pi r_i \Delta H_j K_i}{r_{i+1}-r_{i-1}}T_{i+1,j}^{n+1} = -\frac{\pi\left(r_i^2-r_{i-1}^2\right)\Delta H_j \rho_i C_i}{\Delta t}T_{i,j}^{n+1} \quad \text{(A9)}$$

where $W_j$ represents the heat generated by friction between the fracturing fluid and casing wall, J; $Q$ represents the displacement of fracturing fluid, $m^3$/min, $\rho$ represents density, $kg/m^3$, $C$ represents specific heat, J/(kg·°C), $r$ is the radius, m; $\Delta H_j$ represents the height of the control unit body, m; $U$ represents the convective heat transfer coefficient between the fracturing fluid and casing wall, w/($m^2$·°C); and $\lambda_{fj}$ represents the casing friction coefficient, dimensionless.

During the progress of meshing, when $i = 0$, $1 \leq i < m$, $m \leq i < n$, $n \leq i < o$, $o \leq i < p$, $p \leq i < q$, the meshing grids represent the fracturing fluid, production casing, cement sheath (1), intermediate casing, cement sheat (2), and formation, as show in Figure A1.

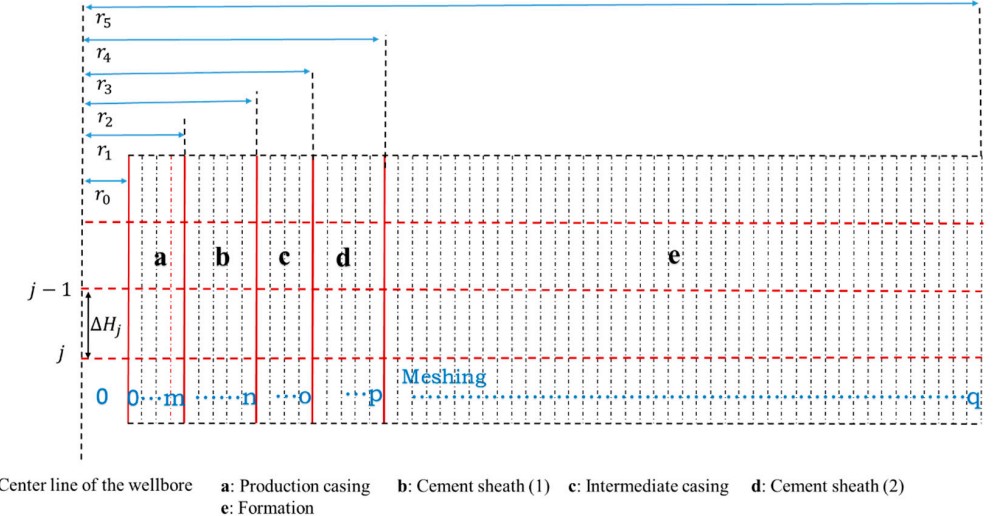

**Figure A1.** Meshing grids of the assembly.

When the fracturing fluid was pumped into the wellbore downhole with high discharge, it was in the state of turbulence, The convective heat transfer coefficient between the casing and fluid can be calculated by using the Marshall model shown in Equation (A10) [34].

$$U = \frac{S_t K_0}{D} = 0.0107\frac{K_0}{D}\left\{\rho_0 D_{eff}\frac{4Q}{\pi D^2} / \left[K\left(\frac{3n+1}{4n}\right)^n \left(\frac{32Q}{\pi D^3}\right)^{n-1}\right]\right\}^{0.67}\left[K_{con}\left(\frac{3n+1}{4n}\right)^n \left(\frac{32Q}{\pi D^3}\right)^{n-1}C_0/K_0\right]^{0.33} \quad \text{(A10)}$$

where $S_t$ is the Stanton number (dimensionless), $K_0$ is the heat conductivity coefficient (w/(m·°C); $D$ is the casing diameter (m); $D_{eff}$ is the equivalent diameter of casing (m); n is the liquidity index, dimensionless; $K_{con}$ is the consistency ($Pa/s^n$); $C_0$ represents the specific heat of the fracturing fluid (J/(kg·°C)).

According to Equations (A5)–(A10), changes in the temperature of the fracturing fluid can be obtained by calculation, and the temperature history of any position along the horizontal section can be expressed as functions that are used as the basic parameters in the finite element model:

$$T_l^Q = g(Q, l, t) \quad \text{(A11)}$$

where $l$ is the distance between the selected position and the toe-end (m).

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
