# Peer review of "Mechanisms and Influence of Casing Shear Deformation near the Casing Shoe, Based on MFC Surveys during Multistage Fracturing in Shale Gas Wells in Canada"

_energies, doi:10.3390/en12030372_

Round 1
Reviewer 1 Report
The paper addresses the issues related to determining casing deformation. The graphics and diagrams show the results of analysis well. The methods related to numerical post-processing are described very shortly in appendix.
The title is misleading - it implies that shear deformations are analyzed, processed and modelled, but, from the text - only the effect of shear stress (and all other kinds of stresses?) is represented.
The workflow is not clear - it can be concluded that the result of MFC tool measurement is represented visually - there is no clear description of surrounding conditions that caused such deformations.
The sentence "Based on the previous research..." (line 67) is not clear. There are not important statements in the sentence.
Line 81: "fracturing fluid flowed along a certain passage into the shale beddings" - this is not clear
Overall language seems to be acceptable, but too many sentences begin with "And".
Line 144 - m3 should be formatted.
New model or anything new is not indicated in paper. Empahasizing the word "new" is obsolete. (e.g. in title, line 251 etc.).
regarding "Geological and mechanical parameters" - the data is too scarce to simulate and predict deformations in such heterogeneous media that is penetrated by well. Moreover, there is no lithological data that confirms that averaged data for vertical in-situ stress, "fracturing fluid friction", Young modulus, specific heat etc. are justified.
line 309 - the statement "Then the casing inner pressure in the interface was about 115 MPa." should be explained (how 115 MPa was obtained).
The structure of paper should be changed, the title "6. Engineering verification" should not be after the results - and the title is repeated.
The list of references should be improved - more fundamental works related to casing deformations and stresses in fractured rock systems should be adequatly connected with the analysis presented in paper.
Author Response
January 19, 2019
Manuscript Number: Energies-415960
Manuscript Title: A new investigation for casing shear deformation based on MFC surveys during multistage fracturing in shale gas wells in Canada
Author: Yan Xi, Jun Li, Gonghui Liu, Jianping Li, Jiwei Jiang
Name of Journal: Energies
Dear Reviewer,
Thank you for your comments concerning our manuscript entitled “A new investigation for casing shear deformation based on MFC surveys during multistage fracturing in shale gas wells in Canada”. We found your comments most helpful and have revised the manuscript accordingly.
The comments raised by you were addressed and the amendments were highlighted in blue in the review report. The manuscript was revised submission with new line numbers in the text, some grammar and spelling errors had also been corrected.
We also responded point by point to each “Comments and Suggestions” as listed below, along with a clear indication of the location of the revision.
Hope these will make it more acceptable for publication.
Sincerely yours,
Xi Yan
19 Jan. 2019
Response to Reviewer #1:
We appreciate for your careful read and thoughtful suggestions/comments about the manuscript, and these do help improve this paper. The paper is mainly changed as the follow:
Response to the Form in the review report
(1)
English language and style
( ) Extensive editing of English language and style required
( ) Moderate English changes required
(x) English language and style are fine/minor spell check required
( ) I don't feel qualified to judge about the English language and style
Response to Reviewer #1:
Thank the reviewer for the comments and your valuable reminder. We have corrected the grammar and spelling errors in the paper.
(2)
Yes | Can be improved | Must be improved | Not applicable | |
Does the introduction provide sufficient background and include all relevant references? | ( ) | (x) | ( ) | ( ) |
Is the research design appropriate? | ( ) | (x) | ( ) | ( ) |
Are the methods adequately described? | ( ) | ( ) | (x) | ( ) |
Are the results clearly presented? | ( ) | ( ) | (x) | ( ) |
Are the conclusions supported by the results? | (x) | ( ) | ( ) | ( ) |
Response to Reviewer #1:
Thank the reviewer for the comments and valuable reminder.
To better respond to the comments of Reviewer#1, we have revised the introduction carefully and optimized the structure of the paper. In addition, we have supplement some details to the Sections of “5. Results” , “6. Results comparison and mitigation method” and “Appendix A & B”, so as to better describe the methods and present the results.
Response to the Comments and Suggestions in the review report
(1) The paper addresses the issues related to
determining casing deformation. The graphics and diagrams show the
results of analysis well. The methods related to numerical
post-processing are described very shortly in appendix.
Response to Reviewer #1 comment No.1:
Many thanks for the reviewer’s nice suggestion. The reviewer’s suggestions have been adopted and we have supplemented more details to the Appendix A & B, so as to make sure the description was clear. Please see Page 16, line 475-485; Page 18, Line 514-518.
(2) The title is misleading - it implies that shear deformations are analyzed, processed and modelled, but, from the text - only the effect of shear stress (and all other kinds of stresses?) is represented.
Response to Reviewer #1 comment No.2:
Thank the reviewer for the comments and your valuable reminder. The objective of this paper was to analyze the reason of fault slipping occurred at the position near the casing shoe and study the influence of fault slipping under different material and mechanical parameters. All things above and the comment No.8 considered, the title of the paper was amended as: “Mechanisms and influences of casing shear deformation near the casing shoe based on MFC surveys during multistage fracturing in shale gas wells in Canada”. Please see Page 1, line 2-5.
(3) The workflow is not clear - it can be concluded that the result of MFC tool measurement is represented visually - there is no clear description of surrounding conditions that caused such deformations.
Response to Reviewer #1 comment No.3:
Thank the reviewer for the comments. To better respond to this comment, we provide specific explanations from two aspects:
1) Description of workflow
Casing shear deformed points occurred at the position of the interface between Nisku and Ireton Formation were all located near the production casing shoes, the mechanism of this kind of casing deformation was introduce and verified in the Section “3. Mechanisms of casing shear deformation induced by multistage fracturing”. MFC tool could be used to detect and measure the deformed part, this kind of technique remains challenging to implement in the extensive regional shale gas field due to its high cost. As a result of this, there was an urgent need of analyzing the mechanism of fault slipping and studying the influence of it on the variation of production casing inner diameter.
Based on the analysis above, the workflow could be expressed by the following flow diagram
2) Description( of the surrounding conditions
First, we have revised the details of the Section “2.1. Field description” and “3.1. Mechanisms of casing shear deformation”, so as to make the description of the surrounding conditions more clear. Please see Page 3, line 135-136; Page 7, line 213-216.
Second, it is worth mentioning that during the progress of analysis, a lot of drilling and completion data, logging data, geological data and microseismic data were collected to analyze the surrounding conditions of the casing deformed part, but most of them were summed up in a few sentences. Take assessing the relationship between the interface between the two formations and casing deformed point as example, nearly all the logging datas of the shale wells were collected and compared, as shown in Fig.2, so as to make sure the locations of all the casing deformed points.
In order to make the description of surrounding conditions in the paper more clear, all of sentences related to this were revised.
3) The sentence “Based on the previous research...” (line 67) is not clear. There are not important statements in the sentence.
Response to Reviewer #1 comment No.3:
Thank the reviewer for the comments. This sentence was optimized aiming to explaine the mechanism of fault slipping during multistage fracturing. Some new references were also supplemented to this sentence. Please see Page2, line 71-74.
The new sentences was as follows
“Based on the previous research work, after fracturing fluid flew into the fractures in the formation, pore pressure in the fractures increased, which caused the normal stress and friction coefficient between the contact surfaces to decrease, then the fault was likely to slip along the fractures under the action of its own gravity or external forces.”
4) Line 81: "fracturing fluid flowed along a certain passage into the shale beddings" - this is not clear
Response to Reviewer #1 comment No.5:
Thank the reviewer for the comments and your valuable reminder. We have rewritten this sentence so as to make it easier to understand. Please see Page 2, line 82-83.
5) Overall language seems to be acceptable, but too many sentences begin with "And".
Response to Reviewer #1 comment No.6:
Thank the reviewer for the comments. The reviewer’s suggestions have been adopted and the full text of the paper was revised, some "And" were optimized away.
6) Line 144 - m3 should be formatted.
Response to Reviewer #1 comment No.7:
Thank the reviewer for the valuable reminder. The " m3/min" was replaced by " m3/min".
Please see Page 4, line 142.
7) New model or anything new is not indicated in paper. Empahasizing the word "new" is obsolete. (e.g. in title, line 251 etc.).
Response to Reviewer #1 comment No.8:
Thank the reviewer for the comments and your valuable reminder. Two changes were conducted:
First, the title was changed from "A new investigation for casing shear deformation based on MFC surveys during multistage fracturing in shale gas wells in Canada" to“Mechanisms and influences of casing shear deformation near the casing shoe based on MFC surveys during multistage fracturing in shale gas wells in Canada”. Please see Page 1, line 2-5.
Second, most of the “new” in the paper (including in the line 251) were optimized away.
8) Regarding "Geological and mechanical parameters" - the data is too scarce to simulate and predict deformations in such heterogeneous media that is penetrated by well. Moreover, there is no lithological data that confirms that averaged data for vertical in-situ stress, "fracturing fluid friction", Young modulus, specific heat etc. are justified.
Response to Reviewer #1 comment No.9:
Thank the reviewer for the comments and your valuable reminder. We have supplemented some details of geological and mechanical parameters to the paper. Please Page 10, line 311.
It's worth explaining that most of the geological and mechanical parameters were obtained from the geological design, logging report and engineering data, the progress of calculation and inversion was complicated. In particular, there were a lot of parameters. As a result of this, only the results were shown in the paper.
Take lithological data as example, as shown in Fig.2, the geological stratification, logging report and MFC data were needed to determine the relationship between the locations of casing deformed part and the depth of the interface.
Another example, we should extract data from the logging report in order to assess the geothermal gradient, as shown in Fig.3. But the progress could not be put in the paper. As a result of this, the Section “Geological and mechanical parameters” seems like insufficient.
All the geological and mechanical parameters such as vertical in-situ stress, fracturing fluid friction, Young modulus and so on, were all from the engineering project, so as to make sure the validity of the calculated results.
9) line 309 - the statement "Then the casing inner pressure in the interface was about 115 MPa." should be explained (how 115 MPa was obtained).
Response to Reviewer #1 comment No.10:
Thank the reviewer for the comments and your valuable reminder. We have supplemented the sentence of “the fracturing fluid density was 1.28g/cm³” to the paper. Please Page 10, line 311.
Then the casing inner pressure in the interface can be calculated as the following:
The pumping pressure of fracturing was 77 MPa. The true vertical depth of the research object was 3742 m, the fracturing fluid density was 1.28g/cm³, then the hydrostatic pressure was 48 MPa. The fracturing fluid friction was approximately 10 MPa, according to the engineering report.
As a result of this, the casing inner pressure = 77 MPa +48 MPa -10 MPa = 115 MPa
10) The structure of paper should be changed, the title "6. Engineering verification" should not be after the results - and the title is repeated.
Response to Reviewer #1 comment No.11:
Thank the reviewer for the comments and your valuable reminder. The title of “6. Engineering verification” was replaced by “6. Results comparison and mitigation method” Please see Page 10, line 411.
11) The list of references should be improved - more fundamental works related to casing deformations and stresses in fractured rock systems should be adequatly connected with the analysis presented in paper.
Response to Reviewer #1 comment No.11:
Thank the reviewer for the valuable reminder. Four related references were supplemented to the paper. Please see Page19, line 561-568, line 576-579; Page 20, line 589-591.

Reviewer 2 Report
The paper is well written and contributes to field data, numerical simulation, and validation. I believe it is appropriate for this journal audience.
Author Response
January 19, 2019
Manuscript Number: Energies-415960
Manuscript Title: A new investigation for casing shear deformation based on MFC surveys during multistage fracturing in shale gas wells in Canada
Author: Yan Xi, Jun Li, Gonghui Liu, Jianping Li, Jiwei Jiang
Name of Journal: Energies
Dear Reviewer,
Thank you for your comments concerning our manuscript entitled “A new investigation for casing shear deformation based on MFC surveys during multistage fracturing in shale gas wells in Canada”. We found your comments most helpful and have revised the manuscript accordingly.
The comments raised by you were addressed and the amendments were highlighted in blue in the review report. The manuscript was revised submission with new line numbers in the text, some grammar and spelling errors had also been corrected.
We also responded point by point to each comments as listed below, along with a clear indication of the location of the revision.
Hope these will make it more acceptable for publication.
Sincerely yours,
Xi Yan
19 Jan. 2019
Response to Reviewer #2:
We appreciate for your careful read and thoughtful suggestions/comments about the manuscript, and these do help improve this paper. The paper is mainly changed as the follow:
Response to the Form in the review report
(1)
English language and style
( ) Extensive editing of English language and style required
( ) Moderate English changes required
(x) English language and style are fine/minor spell check required
( ) I don't feel qualified to judge about the English language and style
Response to Reviewer #1 comment No.1:
Thank the reviewer for the comments and your valuable reminder. We have corrected the grammar and spelling errors in the paper.
(2)
Yes | Can be improved | Must be improved | Not applicable | |
Does the introduction provide sufficient background and include all relevant references? | (x) | ( ) | ( ) | ( ) |
Is the research design appropriate? | (x) | ( ) | ( ) | ( ) |
Are the methods adequately described? | (x) | ( ) | ( ) | ( ) |
Are the results clearly presented? | (x) | ( ) | ( ) | ( ) |
Are the conclusions supported by the results? | (x) | ( ) | ( ) | ( ) |
Response to Reviewer #1 comment No.2:
Thank the reviewer for the comments.
Response to the Comments and Suggestions in the review report
(1) The paper is well written and contributes to field data, numerical simulation, and validation. I believe it is appropriate for this journal audience.
Response to Reviewer #2 comment:
Thank the reviewer for the affirmation and encouragement.

Reviewer 3 Report
This study on casing shear deformation is very relevant for shale gas, extracted by multistage hydraulic fracturing. The finding of this study, related to interface between different formation that opened up due to activation of fault during multistage fracturing and classification of the deformed points based on survey, is also supported by microseismic data. Further, numerical modelling has been used to identify aspects to improve the integrity of the casing.
The work can be improved by revising the introduction that has lots of repetition and clearly addressing the objectives of the study and their relevance to shale gas industry. Relevant geological information and missing references (see attached text) must be provided. Further, the language needs to be worked on to improve the quality of the presentation.
Further comments are provided with the text in the attached file.

Author Response
January 19, 2019
Manuscript Number: Energies-415960
Manuscript Title: A new investigation for casing shear deformation based on MFC surveys during multistage fracturing in shale gas wells in Canada
Author: Yan Xi, Jun Li, Gonghui Liu, Jianping Li, Jiwei Jiang
Name of Journal: Energies
Dear Reviewer,
Thank you for your comments concerning our manuscript entitled “A new investigation for casing shear deformation based on MFC surveys during multistage fracturing in shale gas wells in Canada”. We found your comments most helpful and have revised the manuscript accordingly.
The comments raised by you were addressed and the amendments were highlighted in blue in the review report. The manuscript was revised submission with new line numbers in the text, some grammar and spelling errors had also been corrected.
We also responded point by point to each comments as listed below, along with a clear indication of the location of the revision.
Hope these will make it more acceptable for publication.
Sincerely yours,
Xi Yan
19 Jan. 2019
Response to Reviewer #3:
We appreciate for your careful read and thoughtful suggestions/comments about the manuscript, and these do help improve this paper. The paper is mainly changed as the follow:
Response to the Form in the review report
(1)
English language and style
( ) Extensive editing of English language and style required
(x) Moderate English changes required
( ) English language and style are fine/minor spell check required
( ) I don't feel qualified to judge about the English language and style
Response to Reviewer #3:
Thank the reviewer for your valuable reminder. We have revised the full text and polished the language of the paper.
(2)
Yes | Can be improved | Must be improved | Not applicable | |
Does the introduction provide sufficient background and include all relevant references? | ( ) | ( ) | (x) | ( ) |
Is the research design appropriate? | ( ) | (x) | ( ) | ( ) |
Are the methods adequately described? | (x) | ( ) | ( ) | ( ) |
Are the results clearly presented? | (x) | ( ) | ( ) | ( ) |
Are the conclusions supported by the results? | (x) | ( ) | ( ) | ( ) |
Response to Reviewer #3:
Thank the reviewer for the comments and valuable reminder.
To better respond to the comments of Reviewer#3, we have revised the introduction carefully and supplemented some details to the introduction. Four related references were added to the introduction.
And also, the research design was optimized.
Response to the Comments and Suggestions in the review report
(1) This study on casing shear deformation is very relevant for shale gas, extracted by multistage hydraulic fracturing. The finding of this study, related to interface between different formation that opened up due to activation of fault during multistage fracturing and classification of the deformed points based on survey, is also supported by microseismic data. Further, numerical modelling has been used to identify aspects to improve the integrity of the casing.
Response to Reviewer #3 comment No.1:
Thank the reviewer for the comments and affirmation of the paper.
(2) The work can be improved by revising the introduction that has lots of repetition and clearly addressing the objectives of the study and their relevance to shale gas industry. Relevant geological information and missing references (see attached text) must be provided. Further, the language needs to be worked on to improve the quality of the presentation.
Response to Reviewer #3 comment No.2:
Many thanks for the reviewer’s nice suggestion. The reviewer’s suggestions have been adopted and the paper was revised in the following three aspects:
1) The introduction was revised so as to make the objectives of the study clear. All of the repetitions reminded by the reviewer were checked and optimized item by item.
2) Based on the Attached Text which was provided by the reviewer, the full text of the paper was revised, and the relevant geological information and missing references was supplemented to the paper.
3) In order to improve the quality of the presentation, we have revised the full text and polished the language of the paper.
(3) Further comments are provided with the text in the attached file.
Response to Reviewer #3 comment No.3:
Thank the reviewer for the comments and valuable reminder. The full text of the paper was revised item by item according to the Attached Text. The details are as follows.
1) “In this paper, a new 3D fluid-solid-heat coupling numerical model, in which the elastoplastic constitutive relations of materials were considered and solid-shell elements technique were used, was developed to simulate the progress of fault slipping, the reduction of casing inner diameter along the axis was calculated based on the analysis of MFC surveys.” Too long sentence, makes it difficult to follow. Use of abbreviation (MFC) without prior reference should be avoided.
The sentence was modified so as to make the reader better understand. The revised sentence was
“In this paper, a new 3D finite element model was developed to simulate the progress of fault slipping, taking the fluid-solid-heat coupling effect during fracturing into account. For the purpose of increasing the calculation accuracy, the elastoplastic constitutive relations of materials were considered and the solid-shell elements technique was used. The reduction of casing inner diameter along the axis was calculated and the calculation results were compared with the measurement results of Multi-Finger Caliper surveys. ” Please see Page 1, line 17-22.
2) passon ratio ?
It is a spelling error. We have replaced it by “Poisson ratio”. Please Page 1, line 26
3) “32 wells occurred casing deformations”, “five wells occurred casing deformation”. Faulty sentence construction.
The sentences have been corrected: “Casing deformation occurred in 32 wells”, “Casing deformation occurred in five wells”. Please see Page 2, line 60, 62.
4) “Serious casing deformation occurred during multistage fracturing in shale gas wells in Simonette, Canada.” Source of information?
This information came from a sub-project of the project “Study on failure mechanism and control method of wellbore integrity of shale gas horizontal well” (U1762211) from the Key Program of National Natural Science Foundation of China. China university of petroleum (Beijing) and CNPC Logging Co., LTD were cooperating on this sub-project. All the authors participated in the sub-project and collected the original data together.
5) “......all the deformed points were shear deformation” should be modified to “were due to shear deformation”?
The modification has been completed. Please see Page 2, line 67.
6) “......activated......” should be “reactivated”?
The modification has been completed. Please see Page 2, line 72.
7) “......nature fractures......” should be “natural fractures”?
All of the “nature fractures” in this paper were replaced by“natural fractures”. Please see Page 2, line 73, 76; Page 6, line 193, 197, 199-200.
8) “According to the microseismic data, Bao et al. [6] believed that fault activation during and after hydraulic fracturing could be triggered by different mechanisms, including geostatic stress changed due to the elastic response of the rockmass to hydraulic fracturing or pore-pressure changed due to fluid diffusion along a permeable fault zone.” Repetition. Already mentioned previously.
The sentence was replaced by “Zoback and Snee [18] believed that the high pore pressure generated during hydraulic fracturing operations induced slip on preexisting fractures and faults with a wide range of orientations.” And a new reference (Zoback and Snee, 2018) was added to the paper. Please see Page 2, line 77-79; Page 20, 589-591.
9) “.....the unstable planes were more easily activated, for the reason that the gravity of the fault increased the risk of casing shear deformation [13-15].” Again, these information are repetition. Also, the introduction section needs to be concise and focus should be on the present study emphasizing on well location etc.
The repetition in this sentence was deleted and the whole sentence was optimized. Please see Page 2, line 82-83.
10) “.....the actual the deformation.” The second “the” should be deleted?
The modification has been completed. Please see Page 2, line 90.
11) “Guo et al. [20] developed a numerical model including three parts and the middle part slipped, calculated the influences of slip distance, slip angle and mechanical parameters of cement sheath on casing stress.” Faulty sentence construction.
The modification has been completed. The new sentence was “Guo et al. [20] developed a numerical model and calculated the influences of slip distance, slip angle and mechanical parameters of cement sheath on casing stress.” Please see Page 3, line 104-105.
12) “And the numerical simulation results were verified by comparison with MFC measurement results. Sensitivity analysis was conducted, and the influences of slip distance, casing inner pressure, thickness of production casing and intermediate casing, and mechanical parameters of cement sheath on the variation of casing inner diameter in the deformed part were analyzed.”
Again, this information has already been provided earlier.
The purpose of writing this sentence was to give a brief introduction of the whole paper. According to the comment, the sentences were optimized and condensed. The new sentence was
“The numerical simulation results were verified through the measured data. Six influential factors including the slip distance, casing inner pressure, thickness of production casing and intermediate casing, and mechanical parameters of cement sheath were analyzed.”
Please see Page 3, line 113-115.
13) “16 wells occurred casing deformation during multistage fracturing.” Sentence construction?
The sentences have been corrected: “casing deformation occurred in 16 wells”.
Please see Page 4, line 143.
14) “.....accounted for 75%” should be “75% of the total shear deformed points”.
The modification has been completed. Please see Page 4, line 154.
15) “Therefore, it is very meaningful to clarify the mechanism of the first type casing shear deformation, which is also the aim of this study.” Understanding the mechanism of both the shear deformation would be meaningful. his study focuses on one of them.
We studied both two types of the shear deformation. The mechanisms and numerical models of the two types of deformations were different and the first type was relatively more significant. Therefore, the research results of the first type casing shear deformation were presented in this paper. The relevant studies of the second type have been carried out and will be written into a paper in recent times.
16) “The deformed degree represents the slip distance to some extent, but it cannot describe the reduction of casing inner diameter, therefore it cannot be used to accurately evaluate whether or not the bridge plug could pass through the deformed part of the casing.” Complex sentence.
The sentence was optimized so as to make it easier to read and understand. The revised sentence was
“The deformed degree (Fig. 4(b)) can be used to describe the slip distance to some extent, but can not be used to measure the reduction of casing inner diameter. Therefore, the deformed degree can not be used as the basis of evaluating whether the bridge plug could pass through the deformed part.” Please see Page 5, line 179-182.
17) “This is the most common mechanism of casing shear deformation, and was identified in almost all of the shale gas fields.” References?
The references was supplemented to the sentence. Please see Page 6, line 195.
18) “Shear deformed points near the landing point (type 2, Figure. 6).” Incomplete sentence.
The sentence has been corrected as
“Shear deformed points appeared near the landing point (type 2, Figure. 6).” Please see Page 6, line 202.
19) “Some scholars believed that......” Please provide references.
The references was supplemented to the sentence. Please see Page 7, line 204.
20) “......the minimum increment of pore pressure (MPa) is......”Please number your equations. You need to mention here that the normal stress has been resolved into horizontal and vertical components.
The equations were numbered. Please see Page 7, line 222, 227,231.
We have supplemented the content reminded by the reviewer to the sentence and gave a detailed introduction in Appendix A. Please see Page 16, line 475-485.
21) “In addition, the downhole pressure was as high as 115 MPa, which indicated fault was easy to slip at excessive pore pressure.” Correct sentence please......the downhole pressure was as high as 115 Mpa indicating......
The modification has been completed. Please see Page 7, line 237.
22) “......3742 m and 3786 m.” “ ...... 2.3 MPa/100m, 2.0 MPa/100m, and 2.5 MPa/100m. ” respectively?
The modification has been completed. Please see Page 9, line 307, 309.
23) The pumping pressure of fracturing was 77 MPa, and the fracturing fluid friction was approximately 10 MPa. Then the casing inner pressure in the interface was about 115MPa. Please clarify.
We have supplemented the sentence of “the fracturing fluid density was 1.28g/cm³” to this paragraph. Then the casing inner pressure in the interface can be calculated as the following:
The pumping pressure of fracturing was 77 MPa. The true vertical depth of the research object was 3742 m, the fracturing fluid density was 1.28g/cm³, then the hydrostatic pressure was 48 MPa. The fracturing fluid friction was approximately 10 MPa, according to the engineering report.
As a result of this, the casing inner pressure = 77 MPa +48 MPa -10 MPa = 115 MPa
24) “thckness” Spelling check.
The modification has been completed. “thckness” was replaced by “thickness”. Please see Page 13, line 389.
25) “In addition, according to the mechanism of casing shear deformation occurred at the interface and the sensitivity analysis, decreasing the slip distance was the best way to protect the casing. For the reason that the fracturing plan could not be changed so as to ensure the shale gas production capacity, as a result of this, the well structure was optimized in order to avoid that the interface was opend.” Lacks clarity.
The sentences were checked and optimized so as to achieve the purpose of clear expression. The new sentences were as follows
“According to the above analysis, it could be known that decreasing the slip distance was the best way to protect the casing. In order to mitigate or eliminate fault slipping, the interface between Nisku and Ireton Formation should be avoided being opened during the operation of well cementation. As a consequence, the well structure was optimized.”
Please see Page 15, line 423-426.
26) “......two faults.” Formation?
The modification has been completed. Please see Page 16, line 450.
27) passon ratio ?
We have replaced it with “Poisson ratio”. Please see Page 16, line 455.

Round 2
Reviewer 1 Report
paper is significantly improved after revision, congratulations and good luck.